# Dense Associative Memory for Pattern Recognition

**Dmitry Krotov**
Simons Center for Systems Biology
Institute for Advanced Study
Princeton, USA
krotov@ias.edu

**John J. Hopfield**
Princeton Neuroscience Institute
Princeton University
Princeton, USA
hopfield@princeton.edu

## Abstract

A model of associative memory is studied, which stores and reliably retrieves many more patterns than the number of neurons in the network. We propose a simple duality between this dense associative memory and neural networks commonly used in deep learning. On the associative memory side of this duality, a family of models that smoothly interpolates between two limiting cases can be constructed. One limit is referred to as the feature-matching mode of pattern recognition, and the other one as the prototype regime. On the deep learning side of the duality, this family corresponds to feedforward neural networks with one hidden layer and various activation functions, which transmit the activities of the visible neurons to the hidden layer. This family of activation functions includes logistics, rectified linear units, and rectified polynomials of higher degrees. The proposed duality makes it possible to apply energy-based intuition from associative memory to analyze computational properties of neural networks with unusual activation functions – the higher rectified polynomials which until now have not been used in deep learning. The utility of the dense memories is illustrated for two test cases: the logical gate XOR and the recognition of handwritten digits from the MNIST data set.

## 1   Introduction

Pattern recognition and models of associative memory [1] are closely related. Consider image classification as an example of pattern recognition. In this problem, the network is presented with an image and the task is to label the image. In the case of associative memory the network stores a set of memory vectors. In a typical query the network is presented with an incomplete pattern resembling, but not identical to, one of the stored memories and the task is to recover the full memory. Pixel intensities of the image can be combined together with the label of that image into one vector [2], which will serve as a memory for the associative memory. Then the image itself can be thought of as a partial memory cue. The task of identifying an appropriate label is a subpart of the associative memory reconstruction. There is a limitation in using this idea to do pattern recognition. The standard model of associative memory works well in the limit when the number of stored patterns is much smaller than the number of neurons [1], or equivalently the number of pixels in an image. In order to do pattern recognition with small error rate one would need to store many more memories than the typical number of pixels in the presented images. This is a serious problem. It can be solved by modifying the standard energy function of associative memory, quadratic in interactions between the neurons, by including in it higher order interactions. By properly designing the energy function (or Hamiltonian) for these models with higher order interactions one can store and reliably retrieve many more memories than the number of neurons in the network.

Deep neural networks have proven to be useful for a broad range of problems in machine learning including image classification, speech recognition, object detection, etc. These models are composed of several layers of neurons, so that the output of one layer serves as the input to the next layer. Each

neuron calculates a weighted sum of the inputs and passes the result through a non-linear activation function. Traditionally, deep neural networks used activation functions such as hyperbolic tangents or logistics. Learning the weights in such networks, using a backpropagation algorithm, faced serious problems in the 1980s and 1990s. These issues were largely resolved by introducing unsupervised pre-training, which made it possible to initialize the weights in such a way that the subsequent backpropagation could only gently move boundaries between the classes without destroying the feature detectors [3, 4]. More recently, it was realized that the use of rectified linear units (ReLU) instead of the logistic functions speeds up learning and improves generalization [5, 6, 7]. Rectified linear functions are usually interpreted as firing rates of biological neurons. These rates are equal to zero if the input is below a certain threshold and linearly grow with the input if it is above the threshold. To mimic biology the output should be small or zero if the input is below the threshold, but it is much less clear what the behavior of the activation function should be for inputs exceeding the threshold. Should it grow linearly, sub-linearly, or faster than linearly? How does this choice affect the computational properties of the neural network? Are there other functions that would work even better than the rectified linear units? These questions to the best of our knowledge remain open.

This paper examines these questions through the lens of associative memory. We start by discussing a family of models of associative memory with large capacity. These models use higher order (higher than quadratic) interactions between the neurons in the energy function. The associative memory description is then mapped onto a neural network with one hidden layer and an unusual activation function, related to the Hamiltonian. We show that by varying the power of interaction vertex in the energy function (or equivalently by changing the activation function of the neural network) one can force the model to learn representations of the data either in terms of features or in terms of prototypes.

## 2   Associative memory with large capacity

The standard model of associative memory [1] uses a system of $N$ binary neurons, with values $\pm 1$. A configuration of all the neurons is denoted by a vector $\sigma_i$. The model stores $K$ memories, denoted by $\xi_i^\mu$, which for the moment are also assumed to be binary. The model is defined by an energy function, which is given by

$$E = -\frac{1}{2} \sum_{i,j=1}^{N} \sigma_i T_{ij} \sigma_j, \quad T_{ij} = \sum_{\mu=1}^{K} \xi_i^\mu \xi_j^\mu, \tag{1}$$

and a dynamical update rule that decreases the energy at every update. The basic problem is the following: when presented with a new pattern the network should respond with a stored memory which most closely resembles the input.

There has been a large amount of work in the community of statistical physicists investigating the capacity of this model, which is the maximal number of memories that the network can store and reliably retrieve. It has been demonstrated [1, 8, 9] that in case of random memories this maximal value is of the order of $K^{max} \approx 0.14N$. If one tries to store more patterns, several neighboring memories in the configuration space will merge together producing a ground state of the Hamiltonian (1), which has nothing to do with any of the stored memories. By modifying the Hamiltonian (1) in a way that removes second order correlations between the stored memories, it is possible [10] to improve the capacity to $K^{max} = N$.

The mathematical reason why the model (1) gets confused when many memories are stored is that several memories produce contributions to the energy which are of the same order. In other words the energy decreases too slowly as the pattern approaches a memory in the configuration space. In order to take care of this problem, consider a modification of the standard energy

$$E = -\sum_{\mu=1}^{K} F\left(\xi_i^\mu \sigma_i\right) \tag{2}$$

In this formula $F(x)$ is some smooth function (summation over index $i$ is assumed). The computational capabilities of the model will be illustrated for two cases. First, when $F(x) = x^n$ ($n$ is an integer number), which is referred to as a polynomial energy function. Second, when $F(x)$ is a

rectified polynomial energy function

$$F(x) = \begin{cases} x^n, & x \geq 0 \\ 0, & x < 0 \end{cases} \qquad (3)$$

In the case of the polynomial function with $n = 2$ the network reduces to the standard model of associative memory [1]. If $n > 2$ each term in (2) becomes sharper compared to the $n = 2$ case, thus more memories can be packed into the same configuration space before cross-talk intervenes.

Having defined the energy function one can derive an iterative update rule that leads to decrease of the energy. We use asynchronous updates flipping one unit at a time. The update rule is:

$$\sigma_i^{(t+1)} = Sign\left[ \sum_{\mu=1}^{K} \left( F\left( \xi_i^{\mu} + \sum_{j \neq i} \xi_j^{\mu} \sigma_j^{(t)} \right) - F\left( - \xi_i^{\mu} + \sum_{j \neq i} \xi_j^{\mu} \sigma_j^{(t)} \right) \right) \right], \qquad (4)$$

The argument of the sign function is the difference of two energies. One, for the configuration with all but the $i$-th units clumped to their current states and the $i$-th unit in the "off" state. The other one for a similar configuration, but with the $i$-th unit in the "on" state. This rule means that the system updates a unit, given the states of the rest of the network, in such a way that the energy of the entire configuration decreases. For the case of polynomial energy function a very similar family of models was considered in [11, 12, 13, 14, 15, 16]. The update rule in those models was based on the induced magnetic fields, however, and not on the difference of energies. The two are slightly different due to the presence of self-coupling terms. Throughout this paper we use energy-based update rules.

How many memories can model (4) store and reliably retrieve? Consider the case of random patterns, so that each element of the memories is equal to $\pm 1$ with equal probability. Imagine that the system is initialized in a state equal to one of the memories (pattern number $\mu$). One can derive a stability criterion, i.e. the upper bound on the number of memories such that the network stays in that initial state. Define the energy difference between the initial state and the state with spin $i$ flipped

$$\Delta E = \sum_{\nu=1}^{K} \left( \xi_i^{\nu} \xi_i^{\mu} + \sum_{j \neq i} \xi_j^{\nu} \xi_j^{\mu} \right)^n - \sum_{\nu=1}^{K} \left( - \xi_i^{\nu} \xi_i^{\mu} + \sum_{j \neq i} \xi_j^{\nu} \xi_j^{\mu} \right)^n,$$

where the polynomial energy function is used. This quantity has a mean $\langle \Delta E \rangle = N^n - (N - 2)^n \approx 2nN^{n-1}$, which comes from the term with $\nu = \mu$, and a variance (in the limit of large $N$)

$$\Sigma^2 = \Omega_n (K - 1) N^{n-1}, \qquad \text{where} \quad \Omega_n = 4n^2 (2n - 3)!!$$

The $i$-th bit becomes unstable when the magnitude of the fluctuation exceeds the energy gap $\langle \Delta E \rangle$ and the sign of the fluctuation is opposite to the sign of the energy gap. Thus the probability that the state of a single neuron is unstable (in the limit when both $N$ and $K$ are large, so that the noise is effectively gaussian) is equal to

$$P_{\text{error}} = \int\limits_{\langle \Delta E \rangle}^{\infty} \frac{dx}{\sqrt{2\pi\Sigma^2}} e^{-\frac{x^2}{2\Sigma^2}} \approx \sqrt{\frac{(2n - 3)!!}{2\pi} \frac{K}{N^{n-1}}} e^{-\frac{N^{n-1}}{2K(2n-3)!!}}$$

Requiring that this probability is less than a small value, say $0.5\%$, one can find the upper limit on the number of patterns that the network can store

$$K^{max} = \alpha_n N^{n-1}, \qquad (5)$$

where $\alpha_n$ is a numerical constant, which depends on the (arbitrary) threshold $0.5\%$. The case $n = 2$ corresponds to the standard model of associative memory and gives the well known result $K = 0.14N$. For the perfect recovery of a memory ($P_{\text{error}} < 1/N$) one obtains

$$K^{max}_{\text{no errors}} \approx \frac{1}{2(2n - 3)!!} \frac{N^{n-1}}{\ln(N)} \qquad (6)$$

For higher powers $n$ the capacity rapidly grows with $N$ in a non-linear way, allowing the network to store and reliably retrieve many more patterns than the number of neurons that it has, in accord[1] with [13, 14, 15, 16]. This non-linear scaling relationship between the capacity and the size of the network is the phenomenon that we exploit.

We study a family of models of this kind as a function of $n$. At small $n$ many terms contribute to the sum over $\mu$ in (2) approximately equally. In the limit $n \to \infty$ the dominant contribution to the sum comes from a single memory, which has the largest overlap with the input. It turns out that optimal computation occurs in the intermediate range.

## 3 The case of XOR

The case of XOR is elementary, yet instructive. It is presented here for three reasons. First, it illustrates the construction (2) in this simplest case. Second, it shows that as $n$ increases, the computational capabilities of the network also increase. Third, it provides the simplest example of a situation in which the number of memories is larger than the number of neurons, yet the network works reliably.

The problem is the following: given two inputs $x$ and $y$ produce an output $z$ such that the truth table

| $x$ | $y$ | $z$ |
|-----|-----|-----|
| -1 | -1 | -1 |
| -1 | 1 | 1 |
| 1 | -1 | 1 |
| 1 | 1 | -1 |

is satisfied. We will treat this task as an associative memory problem and will simply embed the four examples of the input-output triplets $x, y, z$ in the memory. Therefore the network has $N = 3$ identical units: two of which will be used for the inputs and one for the output, and $K = 4$ memories $\xi_i^\mu$, which are the four lines of the truth table. Thus, the energy (2) is equal to

$$E_n(x, y, z) = -\big(-x - y - z\big)^n - \big(-x + y + z\big)^n - \big(x - y + z\big)^n - \big(x + y - z\big)^n, \quad (7)$$

where the energy function is chosen to be a polynomial of degree $n$. For odd $n$, energy (7) is an odd function of each of its arguments, $E_n(x, y, -z) = -E_n(x, y, z)$. For even $n$, it is an even function. For $n = 1$ it is equal to zero. Thus, if evaluated on the corners of the cube $x, y, z = \pm 1$, it reduces to

$$E_n(x, y, z) = \begin{cases} 0, & n = 1 \\ C_n, & n = 2, 4, 6, ... \\ C_n xyz, & n = 3, 5, 7, ..., \end{cases} \quad (8)$$

where coefficients $C_n$ denote numerical constants.

In order to solve the XOR problem one can present to the network an "incomplete pattern" of inputs $(x, y)$ and let the output $z$ adjust to minimize the energy of the three-spin configuration, while holding the inputs fixed. The network clearly cannot solve this problem for $n = 1$ and $n = 2$, since the energy does not depend on the spin configuration. The case $n = 2$ is the standard model of associative memory. It can also be thought of as a linear perceptron, and the inability to solve this problem represents the well known statement [17] that linear perceptrons cannot compute XOR without hidden neurons. The case of odd $n \geq 3$ provides an interesting solution. Given two inputs, $x$ and $y$, one can choose the output $z$ that minimizes the energy. This leads to the update rule

$$z = Sign\big[E_n(x, y, -1) - E_n(x, y, +1)\big] = Sign\big[-xy\big]$$

Thus, in this simple case the network is capable of solving the problem for higher odd values of $n$, while it cannot do so for $n = 1$ and $n = 2$. In case of rectified polynomials, a similar construction solves the problem for any $n \geq 2$. The network works well in spite of the fact that $K > N$.

## 4 An example of a pattern recognition problem, the case of MNIST

The MNIST data set is a collection of handwritten digits, which has 60000 training examples and 10000 test images. The goal is to classify the digits into 10 classes. The visible neurons, one for each pixel, are combined together with 10 classification neurons in one vector that defines the state of the network. The visible part of this vector is treated as an "incomplete" pattern and the associative memory is allowed to calculate a completion of that pattern, which is the label of the image.

Dense associative memory (2) is a recurrent network in which every neuron can be updated multiple times. For the purposes of digit classification, however, this model will be used in a very limited

capacity, allowing it to perform only one update of the classification neurons. The network is initialized in the state when the visible units $v_i$ are clamped to the intensities of a given image and the classification neurons are in the off state $x_\alpha = -1$ (see Fig.1A). The network is allowed to make one update of the classification neurons, while keeping the visible units clamped, to produce the output $c_\alpha$. The update rule is similar to (4) except that the sign is replaced by the continuous function $g(x) = \tanh(x)$

$$c_\alpha = g\left[\beta \sum_{\mu=1}^{K}\left(F\left(-\xi_\alpha^\mu x_\alpha + \sum_{\gamma\neq\alpha}\xi_\gamma^\mu x_\gamma + \sum_{i=1}^{N}\xi_i^\mu v_i\right) - F\left(\xi_\alpha^\mu x_\alpha + \sum_{\gamma\neq\alpha}\xi_\gamma^\mu x_\gamma + \sum_{i=1}^{N}\xi_i^\mu v_i\right)\right)\right], \quad (9)$$

where parameter $\beta$ regulates the slope of $g(x)$. The proposed digit class is given by the number of a classification neuron producing the maximal output. Throughout this section the rectified polynomials (3) are used as functions $F$. To learn effective memories for use in pattern classification, an objective function is defined (see Appendix A in Supplemental), which penalizes the discrepancy

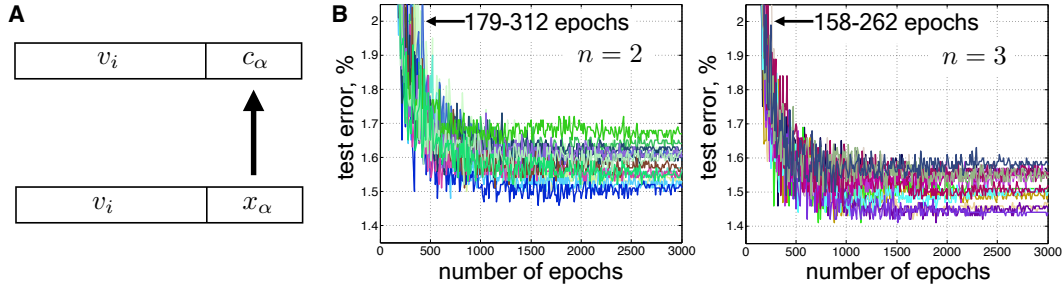

Figure 1: (A) The network has $N = 28 \times 28 = 784$ visible neurons and $N_c = 10$ classification neurons. The visible units are clamped to intensities of pixels (which is mapped on the segment $[-1, 1]$), while the classification neurons are initialized in the state $x_\alpha$ and then updated once to the state $c_\alpha$. (B) Behavior of the error on the test set as training progresses. Each curve corresponds to a different combination of hyperparameters from the optimal window, which was determined on the validation set. The arrows show the first time when the error falls below a 2% threshold. All models have $K = 2000$ memories (hidden units).

between the output $c_\alpha$ and the target output. This objective function is then minimized using a backpropagation algorithm. The learning starts with random memories drawn from a Gaussian distribution. The backpropagation algorithm then finds a collection of $K$ memories $\xi_{i,\alpha}^\mu$, which minimize the classification error on the training set. The memories are normalized to stay within the $-1 \leq \xi_{i,\alpha}^\mu \leq 1$ range, absorbing their overall scale into the definition of the parameter $\beta$.

The performance of the proposed classification framework is studied as a function of the power $n$. The next section shows that a rectified polynomial of power $n$ in the energy function is equivalent to the rectified polynomial of power $n-1$ used as an activation function in a feedforward neural network with one hidden layer of neurons. Currently, the most common choice of activation functions for training deep neural networks is the ReLU, which in our language corresponds to $n = 2$ for the energy function. Although not currently used to train deep networks, the case $n = 3$ would correspond to a rectified parabola as an activation function. We start by comparing the performances of the dense memories in these two cases.

The performance of the network depends on $n$ and on the remaining hyperparameters, thus the hyperparameters should be optimized for each value of $n$. In order to test the variability of performances for various choices of hyperparameters at a given $n$, a window of hyperparameters for which the network works well on the validation set (see the Appendix A in Supplemental) was determined. Then many networks were trained for various choices of the hyperparameters from this window to evaluate the performance on the test set. The test errors as training progresses are shown in Fig.1B. While there is substantial variability among these samples, on average the cluster of trajectories for $n = 3$ achieves better results on the test set than that for $n = 2$. These error rates should be compared with error rates for backpropagation alone without the use of generative pretraining, various kinds of regularizations (for example dropout) or adversarial training, all of which could be added to our construction if necessary. In this class of models the best published results are all[2] in the 1.6% range [18], see also controls in [19, 20]. This agrees with our results for $n = 2$. The $n = 3$ case does slightly better than that as is clear from Fig.1B, with all the samples performing better than 1.6%.

Higher rectified polynomials are also faster in training compared to ReLU. For the $n = 2$ case, the error crosses the $2\%$ threshold for the first time during training in the range of 179-312 epochs. For the $n = 3$ case, this happens earlier on average, between 158-262 epochs. For higher powers $n$ this speed-up is larger. This is not a huge effect for a small dataset such as MNIST. However, this speed-up might be very helpful for training large networks on large datasets, such as ImageNet. A similar effect was reported earlier for the transition between saturating units, such as logistics or hyperbolic tangents, to ReLU [7]. In our family of models that result corresponds to moving from $n = 1$ to $n = 2$.

**Feature to prototype transition**

How does the computation performed by the neural network change as $n$ varies? There are two extreme classes of theories of pattern recognition: feature-matching and formation of a prototype. According to the former, an input is decomposed into a set of features, which are compared with those stored in the memory. The subset of the stored features activated by the presented input is then interpreted as an object. One object has many features; features can also appear in more than one object. The prototype theory provides an alternative approach, in which objects are recognized as a whole. The prototypes do not necessarily match the object exactly, but rather are blurred abstract

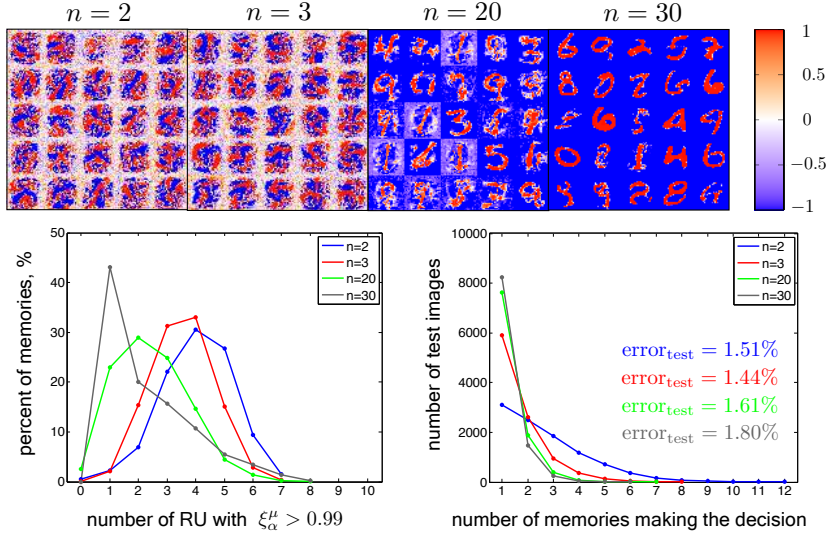

Figure 2: We show 25 randomly selected memories (feature detectors) for four networks, which use rectified polynomials of degrees $n = 2, 3, 20, 30$ as the energy function. The magnitude of a memory element corresponding to each pixel is plotted in the location of that pixel, the color bar explains the color code. The histograms at the bottom are explained in the text. The error rates refer to the particular four samples used in this figure. RU stands for recognition unit.

representations which include all the features that an object has. We argue that the computational models proposed here describe feature-matching mode of pattern recognition for small $n$ and the prototype regime for large $n$. This can be anticipated from the sharpness of contributions that each memory makes to the total energy (2). For large $n$ the function $F(x)$ peaks much more sharply around each memory compared to the case of small $n$. Thus, at large $n$ all the information about a digit must be written in only one memory, while at small $n$ this information can be distributed among several memories. In the case of intermediate $n$ some learned memories behave like features while others behave like prototypes. These two classes of memories work together to model the data in an efficient way.

The feature to prototype transition is clearly seen in memories shown in Fig.2. For $n = 2$ or 3 each memory does not look like a digit, but resembles a pattern of activity that might be useful for recognizing several different digits. For $n = 20$ many of the memories can be recognized as digits, which are surrounded by white margins representing elements of memories having approximately zero values. These margins describe the variability of thicknesses of lines of different training examples and mathematically mean that the energy (2) does not depend on whether the corresponding pixel is on or off. For $n = 30$ most of the memories represent prototypes of whole digits or large portions of digits, with a small admixture of feature memories that do not resemble any digit.

The feature to prototype transition can be visualized by showing the feature detectors in situations when there is a natural ordering of pixels. Such ordering exists in images, for example. In general situations, however, there is no preferred permutation of visible neurons that would reveal this structure (*e.g.* in the case of genomic data). It is therefore useful to develop a measure that permits a distinction to be made between features and prototypes in the absence of such visual space. Towards the end of training most of the recognition connections $\xi_\alpha^\mu$ are approximately equal to $\pm 1$. One can choose an arbitrary cutoff, and count the number of recognition connections that are in the "on" state ($\xi_\alpha^\mu = +1$) for each memory. The distribution function of this number is shown on the left histogram in Fig.2. Intuitively, this quantity corresponds to the number of different digit classes that a particular memory votes for. At small $n$, most of the memories vote for three to five different digit classes, a behavior characteristic of features. As $n$ increases, each memory specializes and votes for only a single class. In the case $n = 30$, for example, more than $40\%$ of memories vote for only one class, a behavior characteristic of prototypes. A second way to see the feature to prototype transition is to look at the number of memories which make large contributions to the classification decision (right histogram in Fig.2). For each test image one can find the memory that makes the largest contribution to the energy gap, which is the sum over $\mu$ in (9). Then one can count the number of memories that contribute to the gap by more than 0.9 of this largest contribution. For small $n$, there are many memories that satisfy this criterion and the distribution function has a long tail. In this regime several memories are cooperating with each other to make a classification decision. For $n = 30$, however, more than 8000 of 10000 test images do not have a single other memory that would make a contribution comparable with the largest one. This result is not sensitive to the arbitrary choice (0.9) of the cutoff. Interestingly, the performance remains competitive even for very large $n \approx 20$ (see Fig.2) in spite of the fact that these networks are doing a very different kind of computation compared with that at small $n$.

## 5   Relationship to a neural network with one hidden layer

In this section we derive a simple duality between the dense associative memory and a feedforward neural network with one layer of hidden neurons. In other words, we show that the same computational model has two very different descriptions: one in terms of associative memory, the other one in terms

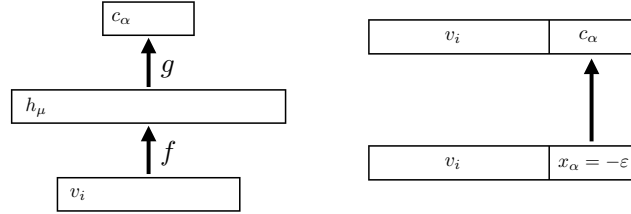

Figure 3: On the left a feedforward neural network with one layer of hidden neurons. The states of the visible units are transformed to the hidden neurons using a non-linear function $f$, the states of the hidden units are transformed to the output layer using a non-linear function $g$. On the right the model of dense associative memory with one step update (9). The two models are equivalent.

of a network with one layer of hidden units. Using this correspondence one can transform the family of dense memories, constructed for different values of power $n$, to the language of models used in deep learning. The resulting neural networks are guaranteed to inherit computational properties of the dense memories such as the feature to prototype transition.

The construction is very similar to (9), except that the classification neurons are initialized in the state when all of them are equal to $-\varepsilon$, see Fig.3. In the limit $\varepsilon \to 0$ one can expand the function $F$ in (9) so that the dominant contribution comes from the term linear in $\varepsilon$. Then

$$c_\alpha \approx g\Big[\beta \sum_{\mu=1}^{K} F'\Big(\sum_{i=1}^{N} \xi_i^\mu v_i\Big)(-2\xi_\alpha^\mu x_\alpha)\Big] = g\Big[\sum_{\mu=1}^{K} \xi_\alpha^\mu F'\big(\xi_i^\mu v_i\big)\Big] = g\Big[\sum_{\mu=1}^{K} \xi_\alpha^\mu f\big(\xi_i^\mu v_i\big)\Big], \quad (10)$$

where the parameter $\beta$ is set to $\beta = 1/(2\varepsilon)$ (summation over the visible index $i$ is assumed). Thus, the model of associative memory with one step update is equivalent to a conventional feedforward neural network with one hidden layer provided that the activation function from the visible layer to the hidden layer is equal to the derivative of the energy function

$$f(x) = F'(x) \tag{11}$$

The visible part of each memory serves as an incoming weight to the hidden layer, and the recognition part of the memory serves as an outgoing weight from the hidden layer. The expansion used in (10) is justified by a condition $\sum_{i=1}^{N} \xi_i^\mu v_i \gg \sum_{\alpha=1}^{N_c} \xi_\alpha^\mu x_\alpha$, which is satisfied for most common problems, and is simply a statement that labels contain far less information than the data itself[3].

From the point of view of associative memory, the dominant contribution shaping the basins of attraction comes from the low energy states. Therefore mathematically it is determined by the asymptotics of the activation function $f(x)$, or the energy function $F(x)$, at $x \to \infty$. Thus different activation functions having similar asymptotics at $x \to \infty$ should fall into the same universality class and should have similar computational properties. In the table below we list some common activation

| activation function | energy function | $n$ |
|---|---|---|
| $f(x) = \tanh(x)$ | $F(x) = \ln\big(\cosh(x)\big) \approx x$, at $x \to \infty$ | 1 |
| $f(x) = $ logistic function | $F(x) = \ln\big(1 + e^x\big) \approx x$, at $x \to \infty$ | 1 |
| $f(x) = $ReLU | $F(x) \sim x^2$, at $x \to \infty$ | 2 |
| $f(x) = \mathrm{ReP}_{n-1}$ | $F(x) = \mathrm{ReP}_n$ | $n$ |

functions used in models of deep learning, their associative memory counterparts and the power $n$ which determines the asymptotic behavior of the energy function at $x \to \infty$.The results of section 4 suggest that for not too large $n$ the speed of learning should improve as $n$ increases. This is consistent with the previous observation that ReLU are faster in training than hyperbolic tangents and logistics [5, 6, 7]. The last row of the table corresponds to rectified polynomials of higher degrees. To the best of our knowledge these activation functions have not been used in neural networks. Our results suggest that for some problems these higher power activation functions should have even better computational properties than the rectified liner units.

## 6 Discussion and conclusions

What is the relationship between the capacity of the dense associative memory, calculated in section 2, and the neural network with one step update that is used for digit classification? Consider the limit of very large $\beta$ in (9), so that the hyperbolic tangent is approximately equal to the sign function, as in (4). In the limit of sufficiently large $n$ the network is operating in the prototype regime. The presented image places the initial state of the network close to a local minimum of energy, which corresponds to one of the prototypes. In most cases the one step update of the classification neurons is sufficient to bring this initial state to the nearest local minimum, thus completing the memory recovery. This is true, however, only if the stored patterns are stable and have basins of attraction around them of at least the size of one neuron flip, which is exactly (in the case of random patterns) the condition given by (6). For correlated patterns the maximal number of stored memories might be different from (6), however it still rapidly increases with increase of $n$. The associative memory with one step update (or the feedforward neural network) is exactly equivalent to the full associative memory with multiple updates in this limit. The calculation with random patterns thus theoretically justifies the expectation of a good performance in the prototype regime.

To summarize, this paper contains three main results. First, it is shown how to use the general framework of associative memory for pattern recognition. Second, a family of models is constructed that can learn representations of the data in terms of features or in terms of prototypes, and that smoothly interpolates between these two extreme regimes by varying the power of interaction vertex. Third, there exists a simple duality between a one step update version of the associative memory model and a feedforward neural network with one layer of hidden units and an unusual activation function. This duality makes it possible to propose a class of activation functions that encourages the network to learn representations of the data with various proportions of features and prototypes. These activation functions can be used in models of deep learning and should be more effective than the standard choices. They allow the networks to train faster. We have also observed an improvement of generalization ability in networks trained with the rectified parabola activation function compared to the ReLU for the case of MNIST. While these ideas were illustrated using the simplest architecture of the neural network with one layer of hidden units, the proposed activation functions can also be used in multilayer architectures. We did not study various regularizations (weight decay, dropout, etc), which can be added to our construction. The performance of the model supplemented with these regularizations, as well as performance on other common benchmarks, will be reported elsewhere.

## Footnotes

[1]The $n$-dependent coefficient in (6) depends on the exact form of the Hamiltonian and the update rule. References [13, 14, 15] do not allow repeated indices in the products over neurons in the energy function, therefore obtain a different coefficient. In [16] the Hamiltonian coincides with ours, but the update rule is different, which, however, results in exactly the same coefficient as in (6).

[2]Although there are better results on pixel permutation invariant task, see for example [19, 20, 21, 22].

[3]A relationshp similar to (11) was discussed in [23, 24] in the context of autoencoders.

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
