[Supplementary Material · DAM_supplemental.pdf]

# Dense Associative Memory for Pattern Recognition

## Dmitry Krotov, John J. Hopfield

## Appendix A. Details of experiments with MNIST.

The networks were trained using stochastic gradient descent with minibatches of a relatively large size, 100 digits of each class, 1000 digits in total. Training was done for 3000 epochs. Initial weights were generated from a Gaussian distribution $N(-0.3, 0.3)$. Momentum ($0.6 \leq p \leq 0.95$) was used to smooth out oscillations of gradients coming from the individual minibatches. The learning rate was decreasing with time according to

$$\varepsilon(t) = \varepsilon_0 f^t, \quad f = 0.998, \tag{12}$$

where $t$ is the number of epoch. Typical values are $0.01 \leq \varepsilon_0 \leq 0.04$. The weights (memories) were updated after each minibatch according to

$$V_I^\mu(t) = p V_I^\mu(t-1) - \partial_{\xi_I^\mu} C$$
$$\xi_I^\mu(t) = \xi_I^\mu(t-1) + \varepsilon \frac{V_I^\mu(t)}{\max_J |V_J^\mu(t)|}, \tag{13}$$

where $t$ is the number of update, $I = (i, \alpha)$ is an index which unites the visible and the classification units. The proposed update in (13) is normalized so that the largest update of the weights for each hidden unit (memory) is equal to $\varepsilon$. This normalization is equivalent to using different learning rates for each individual memory. It prevents the network from getting stuck on a plateau. All weights were constrained to stay within the $-1 \leq \xi_I^\mu \leq 1$ range. Therefore, if after an update some weights exceeded 1, they were truncated to make them equal to 1 (and similarly for $-1$). The slope of the function $g(x)$ in (9) is controlled by the effective temperature $\beta = 1/T^n$, which is measured in "neurons" or "pixels". For large $n$ the temperature can be kept constant throughout the entire training ($500 \leq T \leq 700$). For small $n$ we found useful to start at a high temperature $T_i$, and then linearly decrease it to the final value $T_f$ during the first 200 epochs ($250 \leq T_i \leq 400$, $30 \leq T_f \leq 100$). The temperature stays constant after that. All the models have $K = 2000$ memories (hidden units).

The MNIST dataset contains 60000 training examples, which were randomly split into 50000 training cases and 10000 validation cases. For each hyperparameter a window of values was selected, such that the error on the validation set after 3000 epochs is less than a certain threshold. After that the entire set of 60000 examples was used to train the network (for 3000 epochs) for various values of the hyperparameters from this optimal window to evaluate the performance on the test set. The validation set was not used for early stopping.

The objective function is given by

$$C = \sum_{\substack{\text{training} \\ \text{examples}}} \sum_{\alpha=1}^{N_c} \left( c_\alpha - t_\alpha \right)^{2m}, \tag{14}$$

where $t_\alpha$ is the target output ($t_\alpha = -1$ for the wrong classes and $t_\alpha = +1$ for the correct class). The case $m = 1$ corresponds to the standard quadratic error. For large powers $m$ the function $x^{2m}$ is small for $|x| < 1$ and rapidly grows for $|x| > 1$. Therefore, higher values of $m$ emphasize training examples which produce largest discrepancy with the target output more strongly compared to those examples which are already sufficiently close to the target output. Such emphasis encourages the network to concentrate on correcting mistakes and moving the decision boundary farther away from the barely correct examples rather than on fitting better and better the training examples which have already been easily and correctly classified. Although much of what we discuss is valid for arbitrary value of $m$, including $m = 1$, we found that higher values of $m$ reduce overfitting and improve generalization at least in the limit of large $n$. For small $n$, we used $m = 2, 3, 4$. For $n = 20, 30$, larger values of $m \approx 30$ worked better. We also tried cross-entropy objective function together with softmax output units. The results were worse and are not presented here.

The training can be done both in the associative memory description and in the neural network description. The two are related by the duality of section 5. Below we give the explicit expressions for the update rule (13) for these two methods.

Consider a minibatch of size $M$. In the associative memory framework one can define two $(N + N_c) \times MN_c$ matrices $U_J^{\alpha A}$ and $V_J^{\alpha A}$ (index $A = 1...M$ runs over the training examples of the minibatch, greek indices $\alpha, \gamma = 1...N_c$ run over classification neurons, index $i = 1...N$ runs over visible neurons, indices $I, J = 1...(N + N_c)$ unite all the neurons, visible and classification).

$$U_i^{\alpha A} = v_i^A \qquad V_i^{\alpha A} = v_i^A$$
$$U_\gamma^{\alpha A} = -1 \qquad V_\gamma^{\alpha A} = \begin{cases} +1, & \alpha = \gamma \\ -1, & \alpha \neq \gamma \end{cases}$$

The update rule (9) can then be rewritten as

$$c_\alpha^A = g\Big[ \beta \Big( \sum_{\mu=1}^{K} F_n(\xi_J^\mu V_J^{\alpha A}) - F_n(\xi_J^\mu U_J^{\alpha A}) \Big) \Big],$$

where $F_n(x)$ is the rectified polynomial of power $n$, and summation over index $J$ is assumed. The derivative of the objective function (14) is given by

$$\partial_{\xi_I^\mu} C = (2m\beta n) \sum_{A=1}^{M} \sum_{\alpha=1}^{N_c} \left( c_\alpha^A - t_\alpha^A \right)^{2m-1} \Big[ 1 - \left( c_\alpha^A \right)^2 \Big] \Big[ F_{n-1}(\xi_J^\mu V_J^{\alpha A}) V_I^{\alpha A} - F_{n-1}(\xi_J^\mu U_J^{\alpha A}) U_I^{\alpha A} \Big]$$

The indices $A$ and $\alpha$ can be united in one tensor product index, so that the two sums can be efficiently calculated using matrix-matrix multiplication.

While this way of training the network is most closely related to the theoretical calculations presented in the main text, it is computationally inefficient. The second dimension of the matrices $U$ and $V$ is $N_c$ times larger than the size of the minibatch. This can become problematic if the classification problem involves many classes. For this reason it is computationally easier to train the dense memory in the dual description, which is more closely related to the conventional methods used in deep learning. In this framework, the minibatch matrix $v_i^A$ has $N \times M$ elements. The update rule is

$$c_\alpha^A = g\Big[ \beta \sum_{\mu=1}^{K} \xi_\alpha^\mu f_n(\xi_i^\mu v_i^A) \Big],$$

where $f_n(x)$ is a rectified polynomial of power[3] $n$, and summation over the visible index $i = 1...N$ is assumed. The derivatives of the objective function (14) are given by

$$\partial_{\xi_i^\mu} C = (2m\beta n) \sum_{A=1}^{M} \sum_{\alpha=1}^{N_c} \left( c_\alpha^A - t_\alpha^A \right)^{2m-1} \Big[ 1 - \left( c_\alpha^A \right)^2 \Big] \xi_\alpha^\mu f_{n-1}\big( \xi_j^\mu v_j^A \big) v_i^A$$

$$\partial_{\xi_\alpha^\mu} C = (2m\beta) \sum_{A=1}^{M} \left( c_\alpha^A - t_\alpha^A \right)^{2m-1} \Big[ 1 - \left( c_\alpha^A \right)^2 \Big] f_n\big( \xi_j^\mu v_j^A \big),$$

where summation over the visible index $j$ is assumed. These expressions are very similar to the conventional derivatives used in networks with rectified linear activation functions, but they use power activation functions instead. The minibatch training can be efficiently implemented on GPU.

## Appendix B. Capacity of Dense associative memory.

In section 2 of the main text a theoretical calculation of the capacity for model (4) was presented in the case of power energy functions. In section 5 an intuitive argument (based on the low energy states of the Hamiltonian) was given arguing that the capacities of the models with power energy functions and rectified polynomial energy functions should be very similar. In this appendix we compare the theoretical results of section 2 with numerical simulations and numerically validate the intuitive argument about low energy states.

A random set of $K = 2000$ binary memory vectors was generated in the model with $N = 100$ neurons. A collection of 10000 random initial configurations of binary spins were evolved according

to (4) until convergence. The quality of memory recovery was measured by the overlap between the final configuration of spins $\left(\sigma_i = \sigma_i^{(t \to \infty)}\right)$ and the closest memory, $\max_\mu \left( \sum_{i=1}^N \xi_i^\mu \sigma_i \right)$. If the recovery is perfect, this quantity is equal to $N$; if some of the spins failed to match a memory vector, this quantity is smaller than $N$. In Fig. 4 the histograms of the overlaps are shown for $n = 2, 3, 4$ in case of power and rectified polynomial energy functions. For $n = 2, 3$ the number of memories ($K = 2000$) places the model above the capacity (according to (6), $K_{\text{no errors}}^{max} \approx 11$ for $n = 2$ and $K_{\text{no errors}}^{max} \approx 360$ for $n = 3$). Thus, the model is unable to reconstruct the memories. For $n = 4$, the number of memories is below the capacity ($K_{\text{no errors}}^{max} \approx 7240$), thus the distribution sharply peaks at perfect recovery. For $n \geq 5$ all 10000 samples converge to one of the memories. Qualitatively, this behavior is demonstrated by both power models and rectified models.

Figure 4: The histograms of overlaps for models with $n = 2, 3, 4$ with power energy functions (left) and rectified polynomial energy functions (right). Each histogram has 10000 samples in it.

A family of models with $50 \leq N \leq 200$ and $50 \leq K \leq 1500$ was studied. For each combination of $N$ and $K$ a set of binary memory vectors was generated to make a model of associative memory. After that 1000 random binary initial conditions were evolved according to (4) until convergence. $K_{1/2}$ is the number of memories when half (500) of these samples perfectly converge to one of the memories. In Fig. 5 the $K_{1/2}$ dependence of $N$ is shown for the power and the rectified models with $n = 3$. The solid curve is given by Eq.(6). The results of numerical simulations for the case of power activation functions are consistent with the theoretical calculation (6). The results for the rectified polynomials are a little bit above the theoretical curve, but show similar non-linear behavior.

Figure 5: Scaling behavior of the capacity vs. the number of neurons for $n = 3$ with power and rectified polynomial energy functions. Solid curve is the theoretical result (6).

## Acknowledgments

We thank B. Chazelle, D. Huse, A. Levine, M. Mitchell, R. Monasson, L. Peliti, D. Raskovalov, B. Xue, and all the members of the Simons Center for Systems Biology at IAS for useful discussions. We especially thank Y. Roudi for pointing out the reference [13] to us. The work of DK is supported by Charles L. Brown membership at IAS.

## Footnotes

[3]One should remember that the energy function of power $n$ is dual to the activation function of power $n - 1$. Here, for the sake of notations, we describe the training procedure for general $n$.