[Reviews · NeurIPS 2016]

Reviewer 1

Summary

This paper proposes a modified form of associative memory, in which the number of patterns that can be reliably stored is greater than the number of neurons. (This is achieved by using higher-order interactions in the energy-function of the model.) The authors also show some interesting connections/dualities between a version of inference/recall in the proposed model and feedforward mapping in a particular type of feedforward neural net with one hidden layer. Using this connection, they suggest that different forms of activation function may have fundamentally different efficacies when used in neural networks for certain problem types.

Qualitative Assessment

This paper has good technical quality and the experiments — though very simple — are presented clearly and appear thorough. It would be interesting if the authors had been able to demonstrate storage and recall of a K > N completely distinct dense memories rather than the cases explored here in which the recalled portion of the memory is a subset of low cardinality. To my knowledge, the work here is quite novel and adds some interesting results to our understanding of binary-state associative memory networks and their capacity. Furthermore, the connections between such higher-order networks and different forms of feedforward net also seemed interesting and novel. This paper suggests a number of potentially interesting directions and spin-offs connecting associative memory architectures with feed-forward models. Furthermore, it could also foster interesting new angles to approach our theoretical understanding of neural networks. I believe that this work would be interesting and useful to several sections of the broader NIPS community. In terms of clarity, the paper was very clear and easy to read.

Confidence in this Review

3-Expert (read the paper in detail, know the area, quite certain of my opinion)


Reviewer 2

Summary

The paper studies an associative memory model with polynomial and rectified polynomial Hamiltonians. The results focus on the one-step retrieval case, for which the model is closely related to a standard feedforward neural network. The authors motivate the use of rectified polynomial activation functions experimentally and invoking an increase in capacity argument.

Qualitative Assessment

The theoretical contribution presented in 291—310 is a welcome insight on the computational power of ReLUs. The experimental results for rectified polynomial units reported in figures 2 and 3 are interesting and apparently novel, even in the context of standard feedforward multi-layer networks. Being 291—297 a central point of the paper it should be expanded and better justified. Furthermore, the simple capacity analysis developed in p. 3 for the polynomial energy function is invoked here for the rectified polynomial energy function. This has to be justified. The paper starts from and mostly focuses on the associative memory (Hamiltonian) formulation, but then the findings are restricted to one-step retrieval. Can the authors provide at least some discussion on how to extrapolate the current findings to the case of recurrent network dynamics? Could a toy example relevant to learning be designed to illustrate the model under iterative retrieval? The reference list should be extended and the novel aspects of the work more clearly delineated. There is previous published research on variants of the Hopfield model where the energy function is extended to include higher order terms. In particular, energy functions of the form of Eq. 2 with polynomial F have been studied before, for instance by Abbott and Arian (Phys Rev A, 1987; see also Ref. 4 therein). Such works should be cited. For example, the reported nonlinear increase (98–104) in capacity with polynomial degree has been found before. Regarding the introduction, it should be noted that the reformulation of a classification problem as a partial input completion one is not novel. It is actually a common approach to use similar energy-based models to perform classification. I couldn’t understand the remark on 105–108 on the structuring of the higher order interactions. In fact, how was the number of ‘memories’ K set for the MNIST experiments, and where is it reported? Regarding section 5: can a comment be made on how a network with more than one hidden layer could be obtained starting from an associative memory problem? How would the picture change beyond one-step retrieval? The Appendix would become clearer by including the weight update obtained after differentiating C.

Confidence in this Review

2-Confident (read it all; understood it all reasonably well)


Reviewer 3

Summary

This paper first discussed a model of associative memory that is simple but with larger capability. Then the duality between this model and a feed-forward neural network with one layer of hidden units and a new activation function was presented.

Qualitative Assessment

The storage limitation of memory in associative memory models is addressed in this paper. The duality between dense associative memory and NN with one hidden layer gives a new direction of exploring properties of neural networks. This connection can be a benefit for both fields. Based on the above mentioned contributions in the paper, I strongly recommend to accept this paper.

Confidence in this Review

2-Confident (read it all; understood it all reasonably well)


Reviewer 4

Summary

The paper proposed a class of polynomial energy functions which have the dense associative memory that could be larger than the number of neurons in the network. The author derives a duality between this model and a neural network with one layer of hidden units while the training speed of such model would be faster than the widely-used activation function in the neural networks. The author also shows that the model behaves from feature-matching mode to prototype regime with the increasing of polynomial order. The experiments conducted on logical gate XOR and permutation invariant MNIST indicates the practicability of the model.

Qualitative Assessment

The idea of having polynomial energy functions such that the associative memory can be dense is intriguing. Both the property of duality and the experiments suggest higher order functions may be better than the simple rectified linear units. However, the author states that "pixel permutation invariant task for MNIST are all in the 1.6% range" which is not true. The recent papers Adversarial Network(https://arxiv.org/pdf/1412.6572v3.pdf), Ladder Network(https://arxiv.org/pdf/1507.02672v2.pdf) having better results on this task should be cited. In section 4, "Dense associative memory is a recurrent network" should be stated more clearly, a recurrent network usually refer to the model whose "connections between units form a directed cycle"(wikipedia). Training the model on larger dataset would be more interesting to show the faster computation.

Confidence in this Review

2-Confident (read it all; understood it all reasonably well)


Reviewer 5

Summary

In this paper, a novel model of associative memory is proposed. The proposed model defines a new energy function that captures the higher order correlation between multiple neurons, and make a balance between cooperation and competition of neurons. The proposed model is able to store and reliably retrieve larger amount of neurons than traditional associative memory model. The author also proposes a duality between the new associative memory model and a neural network with one hidden layer. Experiments on MNIST shows the effectiveness of the proposed method.

Qualitative Assessment

The paper is well written and the proposed idea is well presented. The conducted experiments give an idea of how the improved associative memory model works. However, I don't think the results of this work have enough scientific significance. The associative memory model is a relatively old method, and its performance cannot compete with current deep neural networks. The proposed method addresses the problem of limited capacity in associative memory model, but the accuracy on MNIST is not state-of-the-art cannot justify that the associative memory model can be competitive in current standard. The authors show an interesting duality between the associative memory model and a single layer neural network model, but do not show any theoretical advantage of associative memory model over plain neural network. I don't see how the proposed method can be incorporated into a deep learning framework to beat the current deep neural network framework.

Confidence in this Review

1-Less confident (might not have understood significant parts)


Reviewer 6

Summary

This paper proposes an extension over the Hopfield Networks which utilizes polynomial rectified polynomial activation functions for the Hopfield Networks. The authors also show the relationship of this model to a single layer MLP. They have showed some experimental results on MNIST.

Qualitative Assessment

This paper, in general, is well-written. But I would recommend the authors to reconsider the structure of the paper. For example, moving the section 5 before section 4 might be better. The authors showed that for their proposed model, the number of stored memories can be much larger than the number of units in the network. They report reasonable results on MNIST. However, their experiments are only limited to MNIST and it is not convincing enough. The motivation of this paper can be stated more clearly. For example, it is not clear why would one want to use dense associative networks on MNIST classification instead of deep neural networks which should be more powerful. The theoretical results and proofs in this paper are very interesting. The authors were able to train Hopfield nets with higher order polynomials, thus they can store more memories than the number of units. They have also shown that on MNIST their classification results improved from n=2 to n=3 which is quite interesting.

Confidence in this Review

2-Confident (read it all; understood it all reasonably well)